# An Agenda-Setting Account for Psychological Typhoon Eye Effect on Responses to the Outbreak of COVID-19 in Wuhan

**DOI:** 10.3390/ijerph20054350

**Published:** 2023-02-28

**Authors:** Shu-Wen Yang, Ming-Xing Xu, Yi Kuang, Yang Ding, Yu-Xin Lin, Fei Wang, Li-Lin Rao, Rui Zheng, Shu Li

**Affiliations:** 1CAS Key Laboratory of Behavioral Science, Institute of Psychology, Chinese Academy of Sciences, Beijing 100101, China; 2Department of Psychology, University of Chinese Academy of Sciences, Beijing 100049, China; 3School of Transportation, Fujian University of Technology, Fuzhou 350108, China; 4Department of Psychology, Hubei University, Wuhan 430062, China; 5Department of Management and Organizations, Eller College of Management, University of Arizona, Tucson, AZ 85721, USA; 6School of Journalism and Communication, Xiamen University, Xiamen 361005, China; 7Department of Psychology, School of Humanities and Social Sciences, Fuzhou University, Fuzhou 350108, China

**Keywords:** agenda setting, risk perception, psychological typhoon eye effect, risk information proportion

## Abstract

During the outbreak of COVID-19 in Wuhan in 2020, we conducted a nationwide survey of 8170 respondents from 31 provinces/municipalities in China via Sojump to examine the relationship between the distance to respondents’ city of residence from Wuhan and their safety concerns and risk perception of the epidemic that occurred in Wuhan City. We found that (1) the farther (psychologically or physically) people were from Wuhan, the more concerned they were with the safety of the epidemic risk in Wuhan, which we dubbed the psychological typhoon eye (PTE) effect on responses to the outbreak of COVID-19; (2) agenda setting can provide a principled account for such effect: the risk information proportion mediated the PTE effect. The theoretical and managerial implications for the PTE effect and public opinion disposal were discussed, and agenda setting was identified to be responsible for the preventable overestimated risk perception.

## 1. Introduction

On 23 January 2020, two days before the Chinese New Year, Wuhan began its 76-day lockdown in response to the COVID-19 outbreak. Such an unexpected epidemic outbreak and unprecedented quarantine quickly made Wuhan the focus of national attention.

Fear is one of the most fundamental instinctive responses to threats, danger, and risks [1]. It is often assumed that fear diminishes as we move away from danger or risk, which is commonly referred to as the “ripple effect” [2]. However, research has shown that this is not always the case. In fact, evidence from the field indicates the opposite—the level of anxiety and concern is often higher for residents who are farther from the risk center. This phenomenon is known as the “psychological typhoon eye (PTE) effect”, which is named after the relatively calm center of a typhoon. The PTE effect highlights the complex relationship between risk perception and distance, including geographical distance from the epicenter [3,4,5], interpersonal relationship distance from the sufferer [6], and the level of involvement in the threat of danger [7]. It suggests that people may exhibit paradoxical psychological responses to danger.

The PTE effect was first reported and named by Li et al. in their Wenchuan earthquake study. After the 8.0 magnitude earthquake on 12 May 2008 in Wenchuan, 2262 adults, including residents from devastated and non-devastated areas, were surveyed [4]. Participants were asked to indicate (1) the probability that an epidemic disease will be widespread in post-earthquake areas, (2) the times (out of 100 aftershocks) residents in the earthquake areas needed to take safety measures, (3) the number of medical doctors needed for every 1000 residents in the earthquake areas, and (4) the number of psychological workers needed for every 1000 residents in the earthquake areas. Surprisingly, the results revealed that the closer people were to the center of the devastated areas, the less the concern about safety and health residents felt. As the rated devastation level increased, the estimated safety measures needed, the probability of the outbreak of an epidemic, and the numbers of medical and psychological workers needed decreased (as shown in Figure 1). Li et al. conducted two sequential surveys of 5216 residents in non-devastated (Fujian Province and Beijing City) and devastated areas (Sichuan and Gansu Provinces) in September–October 2008 and April–May 2009 [6]. They asked participants to respond to the same questions as in the first survey wave and found the same PTE effect, indicating its robustness. For a detailed review of Li et al. sequential surveys, see also [8,9,10,11,12].

The PTE effect has been observed not only in the context of natural hazards but also in the context of terrorist attacks in Xinjiang. Li et al. conducted a survey of 2034 residents from 31 provinces/municipalities in China in 2018 to examine the relationship between the distance to respondents’ city of residence from Ürümqi and their concerns about safety and security concerning the China–Eurasia Expo held in Ürümqi. They found that the closer the respondents lived to Ürümqi, the less concerned they were with the safety and security of the expo, and those who live in Ürümqi had the least concern, as if they were in the peaceful typhoon eye (Figure 2). This new discovery is dubbed the PTE effect in response to terrorism [13].

Corroborating evidence from other studies supports the robustness of such a PTE effect by applying a variety of measures [9]. For example, Xie et al. measured the anxiety level of people in and out of SARS-affected areas [14]; Hoven et al. conducted a post-9/11 study and assessed the psychopathology problems of children at schools far from Ground Zero and from the nearest area [15]; Wang et al. described the ratings of the earthquake-related PTSD of residents who live 0.5 and 10 km away from the epicenter [16]. These observations show a common result that people’s risk perceptions toward hazards follow the model suggested by the PTE effect.

In the same vein, we reasoned that the PTE effect is likely detected in the context of the perception of risk posed by COVID-19 in Wuhan City. The PTE effect over COVID-19 should be interesting and counterintuitive (e.g., [17,18]). Most importantly, such an unexpected epidemic outbreak and unprecedented quarantine presented us with an opportunity to understand and specify a missingness mechanism behind the PTE effect. Almost since the effect was detected in the study of Wenchuan earthquake, the mechanism involved remains unclear, and we do not know what makes it work and what makes it appear repeatedly in different risk domains. A careful review shows that all reported PTE effects have one thing in common: all types of media provide extensive and focused reports to the negative consequences of events after the occurrence of major emergencies. We speculated that this kind of media report makes the people far away from the epicenter aware of the risks posed, but it also serves as an unexpected side effect in shaping public risk perception—the agenda setting of media inadvertently leads to “focusing illusion” [19], which may be the mechanism behind the formation of the PTE effect.

Agenda setting refers to the idea that a strong correlation exists between the emphasis that mass media place on certain issues (e.g., based on relative placement or amount of coverage) and the importance attributed to these issues by mass audiences [20,21]. In the context of major emergencies, “agenda setting” determines the “risk information proportion” of audiences, which we defined as the ratio of “the amount of information related to the occurrence of risk events in a certain area” and “the total amount of information about all events in a certain area.” The total information of all events in a certain area = the relevant information of risk events in a certain area + the information of other events in a certain area (Figure 3).

Agenda setting disproportionately enlarges or amplifies the risk information proportion of audiences, leading amplified risk information to become a focused one. According to focusing illusion theory, such a function leads to “an exaggeration of the importance of ideas that are currently on the agenda” [22]. For example, Schkade and Kahneman asked subjects to indicate the happiness of paraplegics and found that respondents, who said they had never known a paraplegic, estimated a predominance of bad moods over good, whereas those who had known a paraplegic as a friend or relative had the opposite perception [16]. According to focus illusion (or risk information proportion), those who do not know paraplegics typically focus narrowly on miserable facts and thus tend to negatively predict their emotional status. This condition is where the “risk information proportion” may follow and apply to explain the PTE effect.

In the case where Wuhan and even the whole country declared a lockdown, the role of media became more prominent than before. With social and outdoor behaviors being strictly forbidden, many people had to rely on media reports heavily to obtain information about the process of the epidemic risk. We speculated that having been shaped by media agenda setting, the information received by people far from Wuhan was mostly risk-related information (i.e., risk information proportion was large), whereas the one received by people close to Wuhan was not risk-related only but other non-risk ones, including their own experiences (i.e., risk information proportion was small). Information that accounts for a large risk information proportion can easily receive further attention because of focusing illusion. As a result, people residing far from Wuhan focused on risk-related information and weighed it heavily, and then eventually estimated greater risk than Wuhan residents, thereby confirming the pattern described by the PTE effect on responses to the COVID-19 outbreak in Wuhan City.

That is, the distorted risk information proportion perceived by audiences may be the reason for the PTE effect. We therefore proposed our working hypotheses as follows: 

**Hypothesis 1 (H1).** 
*The distance to respondents’ city of residence from Wuhan and their levels of concern for health safety regarding COVID-19 in Wuhan City are positively related. That is, the closer the respondents lived to Wuhan, the less concerned they were with the health safety regarding COVID-19 in Wuhan City.*


**Hypothesis 2 (H2).** 
*Risk information proportion mediates the relationship between the distance to respondents’ city of residence from Wuhan and their safety concerns and risk perception of the epidemic that occurred in Wuhan City.*


## 2. Materials and Methods

### 2.1. Participants

To test the hypotheses and answer the research question, a purposive online survey was conducted with adults as respondents from 31 provinces/municipalities of China, with the exception of Hong Kong, Macao, and Taiwan. From 2 February to 12 February 2020, 8170 participants (56.5% females; M_age_ = 25.50 years, standard deviation (SD) = 10.38) were recruited via Sojump (https://www.wjx.cn/) to complete a 26-item questionnaire. Written informed consent was obtained from all study respondents. The detailed demographic information on the respondents is presented in Table 1. 

#### 2.1.1. Predictor Variables: Distances to Wuhan

This concept was operationalized as subjective (psychological) distance and spatial (physical) distance. 

**Subjective (psychological) distance to Wuhan.** Respondents were asked to rate, on a scale from 0 to 100, how far away their residence is from Wuhan. The closer the rating is to 100, the farther away they perceive themselves to be from Wuhan.

**Spatial (physical) distance to Wuhan.** Respondents were asked to report their residence city when completing the questionnaire. We calculated the straight-line distance on a map from Wuhan to their residence according to the longitude and latitude of the reported location by using the DataMap for Excel (Version 5.1.2, developed by Forrest Studio, Foshan, China). The log transformation of distance was performed for the final analysis.

#### 2.1.2. Outcome Variables: Risk Perception of COVID-19 in Wuhan

We developed 15 items to evaluate the risk perception of COVID-19 in Wuhan. The list of questions included in the survey can be found within the Appendix.

#### 2.1.3. Mediator Variables: Risk Information Proportion

On the basis of the logic illustrated in Figure 3, we designed two measures to estimate the mediator variable (risk information proportion).

The first is a double-item measurement of the risk information proportion: Risk Information Proportion Score I (RIP Score I). Two items on a nine-point Likert scale were measured by asking (1) to what extent do you obtain information about COVID-19 in Wuhan from media (or word of mouth); (2) to what extent do you obtain information about COVID-19 in Wuhan from your firsthand experience. RIP Score I was formulated as follows, in which the higher the score, the more risk information proportion:$$\mathrm{RIP}\;\mathrm{Score}\;I=\frac{\mathrm{score}\;\mathrm{of}\;\mathrm{media}\;s\left(\mathrm{or}\;\mathrm{word}-\mathrm{of}-\mathrm{mouth}\right)}{\mathrm{score}\;\mathrm{of}\;\mathrm{media}+\mathrm{score}\;\mathrm{of}\;\mathrm{experience}}$$

The second is a single-item measurement of the risk information proportion: Risk Information Proportion Score II (RIP Score II). The question was also ranged from 1 (absolutely obtained from firsthand experience) to 9 (absolutely obtained from media (or word of mouth)).

#### 2.1.4. Control Variables

Classical demographical data were collected, including gender, age, and identity. All participants were asked to rate, on a scale from 0 to 100, how important and close Wuhan is to themselves. The closer the rating is to 100, the more important Wuhan is to the participants and the closer they are to Wuhan.

Furthermore, considering that “psychological immunization” may be the potential mechanism for the original version of the PTE effect [6], “time exposed to news about COVID-19 in Wuhan” was considered the indicator for “psychological immunization” and was controlled in the mediation analyses.

### 2.2. Statistical Analyses

#### 2.2.1. Data Preprocessing

First, respondents with ages from 18 to 100 were considered for subsequent analysis (*N* = 7927). For the outcome variables, after the recoding of reverse items (Items 5 and 7), we examined whether the questions were non-responded, and the rating scores were beyond the required range. Second, extreme outliers were identified by boxplot with the use of “outer fences” [23,24]. Last, non-responded questions, incompetent answers, and extreme outliers were all marked as missing values. The range of missing values was from 3.1 to 7.7 for respondents of *N* = 7927. To synthesize the information contained in the 15 items and reduce the dimensionality, we performed an exploratory factor analysis (EFA). We aimed to use the overall scores of the extracted factors to construct an overall index for the risk perception of COVID-19 in Wuhan.

#### 2.2.2. Construct Validity and Reliability

To explore the construct of questionnaire for the risk perception of COVID-19 in Wuhan, Horn’s parallel analysis and EFA (via principal axis factoring (PAF)) with oblique rotation were conducted to test the construct validity with JASP 0.12. Moreover, Cronbach’s coefficient alpha (α) and McDonald’s coefficient omega (ω) were computed to evaluate internal consistency reliability [25,26].

The parallel analysis alongside scree plot [27], cumulative % of variance explained, and interpretability of components were the criteria used for determining the number of factors. The exclusion criteria were to delete (1) items with factor loadings below .50, because loadings ±.50 or greater are considered practically significant and (2) items that cross-loaded on multiple factors with loadings greater than .40 [28].

#### 2.2.3. Analyses

Correlation coefficients were computed for all variables, including predictor, mediator, outcome, and control variables, using complete adult data sets with no missing values. When conducting correlation and regression analyses, one missing value was detected in Lg spatial (physical). As a result, the sample size for the calculation of correlations and regression analyses involving Lg spatial (physical) was adjusted to 7283. Hierarchical multiple regression analyses were conducted to evaluate the effects of distances on the risk perception of COVID-19 in Wuhan. To test whether RIP Score I and RIP Score II functioned as mediators between distances and the risk perception of COVID-19 in Wuhan, we conducted multiple mediation (on the basis of 5000 resamples, Model 4) by using Hayes’ PROCESS macro for SPSS [29], which permits the assessment of multiple indirect effects simultaneously. Our analytic approach was informed by Preacher and Hayes who recommended bias-corrected bootstrapping to measure multiple indirect effects [30]. The totality of all the estimated indirect effects permits the construction of a 95% confidence interval (CI) for the effect size of each indirect effect. If the values of the estimated effect sizes within the CI include zero, then a nonsignificant effect is indicated. All the intervals described in this study were bias corrected. All *p* values reported were two-sided, and *p* values < .05 were considered statistically significant.

## 3. Results

### 3.1. Construct Validity and Reliability

The results of the Kaiser-Meyer-Olkin (KMO) and Bartlett’s test of sphericity showed that the data were suitable for factor analysis (KMO = 0.701, χ^2^ = 15579.744 [*df* = 36, *p* < .001]). Four factors, which had eigenvalues exceeding those randomly generated, and nine items were retained. The four-factor solution explained 48.742% of the total variance. The results suggested that Factor 1 represented the dimension “personal prevention,” Factor 2 the dimension “healthcare,” Factor 3 the dimension “environmental isolation,” and Factor 4 the dimension “interpersonal prevention.” The nine-item scale showed acceptant internal consistency (α = .555; ω = .614). Table 2 presents the items retained and factor loadings.

In the present study, the standardized scores of nine items were used to calculate an overall index for the risk perception of COVID-19 in Wuhan. If any case, had at least one missing value within the nine items, then the case would be excluded from the subsequent analysis. The greater the overall score of the nine items, the more concerned respondents were with the safety regarding the epidemic risk in Wuhan.

### 3.2. Correlation

Pearson’s product–moment correlations were run to examine bivariate relationships (Table 3). The correlations of distances, risk information proportion, and risk perception of COVID-19 in Wuhan showed significance (*p* < .05).

### 3.3. Effect of Distances on the Risk Perception of COVID-19 in Wuhan

Hierarchical ordinary least squares (OLS) regression was used to test H1 for each of the two distance measures. For each analysis, demographic variables, including gender, age, identity, importance, and closeness, were entered as predictors for the first step of the regression model. In the next step, distance was entered together with these demographic variables to determine the effects on the measure, in addition to the controls.

We first analyzed the effect of subjective (psychological) distance on the risk perception of COVID-19 in Wuhan. Overall, with all variables entered, the model explained 4.5% of the variance in having a high risk perception of COVID-19 in Wuhan. In Model 1, demographic variables were entered as predictors, the overall model was significant, *R*^2^ = .034, *F* (5, 7278) = 51.562, and *p* < .001 (Table 4 and Figure 4). In Model 2, subjective (psychological) distance was entered as a predictor, the overall model remained significant, *R*^2^ = .045, *F* (6, 7277) = 56.914, and *p* < .001. Subjective (psychological) distance was a significant predictor of risk perception (β = .106, *p* < .001).

We conducted the same regression analysis to analyze the effect of spatial (physical) distance on the risk perception of COVID-19 in Wuhan. Overall, with all variables entered, the model explained 3.6% of the variance in having a high risk perception of COVID-19 in Wuhan. In Model 1, demographic variables were entered as predictors, the overall model was significant, *R*^2^ = .034, *F* (5, 7277) = 51.572, and *p* < .001 (Table 5 and Figure 5). In Model 2, spatial (physical) distance was entered as a predictor, the overall model remained significant, *R*^2^ = .036, *F* (6, 7276) = 44.781, and *p* < .001. Spatial (physical) distance was a significant predictor of risk perception (β = .038, *p* = .001).

### 3.4. Effect of Distances on the Risk Perception of COVID-19 in Wuhan through Risk Information Proportion

H2 predicted that distances will lead directly to risk information proportion and the risk perception of COVID-19 in Wuhan, and indirectly to risk perception mediated through risk information proportion. We conducted two mediation analyses of subjective (psychological) distance and spatial (physical) distance on risk perception separately.

First, the subjective (psychological) distance was entered as the predictor, RIP Score I and RIP Score II were treated as parallel mediators, and the risk perception of COVID-19 in Wuhan was the outcome or dependent variable. RIP Score I and RIP Score II were treated as parallel mediators because we wanted to test mediations’ alternate form reliabilities in different measurements. Apart from the demographic variables used in regression, time exposed to news about COVID-19 was also employed as a control variable. The reason why “time exposed to news about COVID-19” was underscored as a control variable was because the discussion from previous research suggests that psychological immunization may be an alternative explanation for the PTE effect. As illustrated in Figure 6 and Table 6, people who rated farther away from Wuhan likely received more news about COVID-19 in Wuhan (β 1 = .177, *p* < .001; β 2 = .146, *p* < .001) and had a higher risk perception of COVID-19 in Wuhan (β = .089, *p* < .001), both types of measurement of risk information proportion showed direct effects on the risk perception of COVID-19 in Wuhan (β 1 = .066, *p* < .001; β 2 = .044, *p* < .001). In addition, subjective (psychological) distance exhibited an indirect effect on the risk perception of COVID-19 in Wuhan through both types of measurement of risk information proportion (β 1 = .012, 95% CI = [.007, .017]; β 2 = .006, 95% CI = [.003, .011]), as the bias-corrected bootstrap CI for the indirect effects were entirely above zero, which constitutes statistically significant mediation effects.

Although evidence supported the two mediators, one mediator can account for significantly more variance than the other. To determine the relative value of the mediators, we conducted bias-corrected comparisons between the two mediators. The 95% CI for the contrasts of RIP Score I with RIP Score II included zero (95% CI = [−.002, .013]), indicating that the contrasts of both mediators were insignificant.

Second, we conducted the same analysis of spatial (physical) distance. The results were similar to subjective (psychological) distance, except that spatial (physical) distance had no direct effect on the risk perception of COVID-19 in Wuhan (β = .021, *p* = .069). Spatial (physical) distance showed a direct effect on RIP Score I and RIP Score II (β 1 = .146, *p* < .001; β 2 = .134, *p* < .001). Both mediators had direct effects on the risk perception of COVID-19 in Wuhan (β 1 = .076, *p* < .001; β 2 = .049, *p* < .001), and spatial (physical) distance showed an indirect effect on risk perception through RIP Score I and RIP Score II (β 1 = .011, 95% CI = [.007, .016]; β 2 = .007, 95% CI = [.003, .011]) (Figure 7 and Table 7). The 95% CI for the contrasts of RIP Score I with RIP Score II included zero (95% CI = [−.002, .011]), suggesting that the contrasts of both mediators were insignificant.

## 4. Discussion

During the COVID-19 outbreak in China, we conducted a nationwide survey to investigate safety concerns and risk perception regarding the epidemic in Wuhan. It is noteworthy that the risk source in current PTE effect differs significantly from that of other events such as the Wenchuan earthquake [4], the Xinjiang terrorism [13], and the 9/11 attack [10]. While these events had a clear and single center of risk source, the epidemic’s risk source was multi-source and multi-point. Although the epidemic risk in Wuhan was far from a ripple caused by “one stone thrown into the water at a time” (c.f., [2]), and the outbreak in Wuhan and the spreading to all parts of the 12 country was similar to “multiple big stones thrown into the water many times”, we adopted a consistent approach to assess people’s levels of concern regarding the Wuhan epidemic, as opposed to measuring concern pertaining to epidemic risks in the surrounding areas (e.g., [31,32]) or from the perspective of an “actor” (i.e., evaluating participants’ own psychological states [18]). We found that the PTE effect appears to hold for COVID-19, indicating that participants staying far away from Wuhan City exhibited more safety concerns or fears about the Wuhan epidemic than participants staying close to Wuhan City in China. Therefore, our results must be different from the those using an approach of measuring people’s concern about the epidemic risk around or in other cities.

While it is not known whether our findings can be generalized to other cultures or countries, encouragingly, a small-scale survey conducted by Xu et al. [33] provided supportive evidence for external validity. They surveyed 353 adults from 19 countries via WeChat and found that the PTE effect was applicable to COVID-19. Participants staying abroad showed more safety concerns or fears regarding the epidemic in Wuhan compared with those staying in China. People at zero distance to Wuhan were at the center of the psychological distance effect, and their safety concerns and risk perception were the lowest. However, given that the sampling of this survey was not rigorous enough, a more rigorous cross-national or cross-cultural comparative study is still necessary.

The mediation analysis provided evidence that agenda setting can provide a principled account for the PTE effect. Specifically, the epidemic in Wuhan not only affected the risk perception of local people but also that of people outside Wuhan through the wide spread of media. The closer the people are to Wuhan, the more their perception of Wuhan includes not only the news reports about the epidemic risk but also the information obtained from other channels, including their personal feelings. That is, for Wuhan residents, the epidemic risk information only accounts for a relatively small proportion in all information related to Wuhan. Therefore, local people’s perception of the epidemic risk in Wuhan, corresponding to its small proportion, presents a relatively low level. Conversely, the information channel of people far away from Wuhan is monotonous and fully dominated by news reports about the epidemic risk. Therefore, the public perception of the epidemic risk in Wuhan corresponds to the large proportion of epidemic information, presenting a relatively high level. Moreover, the usually overlooked difference between people close to and far from Wuhan is that the proportion of epidemic information in all information related to Wuhan is different.

The large proportion of information is more likely given attention, and the weight of the impact of such an information on the risk judgment of the place where an incident happened is overestimated, making the risk awareness of the same risk source higher and finally leads to the PTE effect.

According to the account of focusing illusion, attention is likely paid to the information with a large proportion, and the risk perception posed by a large proportion of information is overestimated, leading to the PTE effect. Taking the Australian bushfire in late 2019 (Figure 8, Panels A and B) as an example, people who did not visit Sydney during this period would think that the Sydney bushfire was dangerous, whereas those who visited Sydney during this period would think that the Sydney bushfire was not so dangerous. The reason is that what people outside Sydney knew about the city were all information related to the bushfire, whereas those in Sydney could see and hear information that was neither completely related to nor completely irrelevant to the bushfire. For all mentioned above, we may reach the conclusion that “risk information proportion” should be responsible for the preventable overestimated risk perception.

Recently, the PTE effect has been detected in many risk areas. For major emergencies that have been “touched” by the media, the public risk perception of the risk source is almost the same as what PTE effect describes. That is, the PTE effect found in different risk areas may share a common or domain-general mechanism. However, the potential explanations proposed so far for the PTE effect are domain specific. For example, to provide a possible explanation of the PTE effect in the context of earthquake, cognitive dissonance [34] or “psychological immunization” [35], has been proposed [33]; to account for the PTE effect in the context of lead–zinc pollution, “benefit account” has been developed [7]. Wen et al. investigated the PTE effect in the context of the COVID-19 outbreak from a prospective of “actors versus bystanders” [18]. By contrast, the potential mechanism identified in the present study—the focusing illusion caused by media’s agenda setting—is likely domain general, which is worthy of “once and for all” investigations in various major emergencies in the future.

The understanding of the underlying mechanism of the PTE effect raises an important question, which must arouse our high vigilance: all the PTE effects detected so far may be attributed to the imbalance of media reports. Once emergencies occur, ranging from the controversy caused by a Guangdong–Hong Kong Cup football match to the Black Lives Matter movement and the spread of the epidemic around the world, extensive media reports follow, allowing people spatially far away from the “epicenter” to also pay great attention to the risk source.

However, overwhelming reports about a single focus and limited unrelative information to check and balance can inevitably deviate the attention of onlookers and conduct their overall risk evaluation on the basis of single-faceted information, which consequentially causes the PTE effect. That is, *PTE does not exist in the world, but excessive risk reports make it PTE.* Drawing on the words of Lu Xun, a prominent Chinese writer of the 20th century, the phrase “In fact, the earth had no roads to begin with, but when many men pass one way, a road is made” may be aptly applied to the phenomenon of PTE in the realm of risk perception. Specifically, it suggests that in fact, the earth had no PTE to begin with, but when many men reported the risk in a focusing illusion way, a PTE is made. This implies that the PTE effect is not a natural phenomenon, but a product of human perception. If people rely solely on their own perceptions rather than media reports, their risk perception of risk source (epicenter) will be an unbiased risk perception, so the so-called “psychological typhoon eye effect” will no longer exist. There are roughly two groups of people who can generate unbiased risk perception: (1) people at the center of risk source; (2) people who are not in the center of the risk source, but have experience to learn from such risks, such as the risk perception on the Wenchuan (the epicenter) earthquake (2008) by people who have experienced the Tangshan (the epicenter) earthquake (1976); or the risk perception on the epidemic in Wuhan by the people who have experienced the epidemic in their cities where the epidemic has also occurred or spread.

To mitigate the impact of PTE, the issue of the unbalanced reporting of emergencies (either intentionally or unintentionally) must be thoroughly resolved. Otherwise, every major emergency in the future will inevitably be accompanied by the PTE phenomenon.

Balancing agenda setting is easier to know than to do [36]. To do so, we must adhere to the correct reporting principle—what to be reported is neither completely irrelevant to nor completely related to major emergencies. Specifically, when reporting a local major emergency, we should also report the news unrelated to the major emergency, such as news describing the daily working life of local people, so that people in other areas can see a full picture that local people actually see.

However, the practical implementation of balanced agenda setting now appears to be constitutionally impossible. First, the physical nature of traditional media itself limits the possibility of balanced news reporting. For example, the visual presentation of newspapers is limited by the layout of printed sheets; the auditory presentation of broadcasting and television is limited by scheduled time; and visually or orally reported emergencies are affected by the recency effect and other factors [37]. Fortunately, the development of the Internet and multimedia offers some possibility to solve this problem. For example, for reports on television screens, we may scale down the size of focus events in a whole picture with the picture-in-picture technology (Figure 8). Such a technology grants audiences a global perspective to see emergencies as a small part of what is going on around them and thus reduces focusing illusion. 

Second, the deliberated selection bias of media makes “balanced reporting” difficult to achieve. Profit-oriented media, which desperately attempt to attract additional web traffic and public attention, do not miss any opportunity to intensively report top news about ongoing emergencies. The media influenced by different political views even selectively report the risk information related to major emergencies. Therefore, balancing agenda setting in the news media is a vision of “easier said than done”.

In sum, in the face of the PTE effect associated with major emergencies, using “balanced” media reports to mitigate the impact of the effect is more challenging than detecting the PTE effect in various risk domains. Therefore, setting a “national standard for regulating the way to release information on public health emergencies” and incorporating it into the system of “the emergency plans and the guide to action” are urgently needed to improve our systems for the management of public health emergencies.

Furthermore, it is worth noting that, regarding the risk source at the epicenter, there are at least two possible explanations for the phenomenon of low safety concerns and risk perception among residents. The first is the agenda-setting account, which suggests that the residents in Wuhan were exposed to information that was neither completely related to nor completely irrelevant to the outbreak of COVID-19, resulting in a less biased and hence lower perception of risk. The second explanation is the ‘corridor of attention’ account, which posits that the residents near the epicenter were experiencing high stress and had dulled emotional responses, leading to a decreased sensitivity to the outbreak of COVID-19. It is our hope that future field studies will provide data to determine which of these two competing accounts is superior and thus preferable.

## 5. Conclusions

During the Spring Festival holidays of 2020, we conducted a nationwide survey of 8170 respondents from 31 provinces/municipalities in China via Sojump to examine the relation between the distance to respondents’ city of residence from Wuhan and their safety concerns and risk perception of the epidemic that occurred in Wuhan City. We found that the farther (psychologically or physically) people were from Wuhan, the more concerned they were with the safety regarding the epidemic risk in Wuhan. People at zero distance to Wuhan were at the center of the PTE, and their safety concerns and risk perception were the lowest. We dubbed it the PTE effect in responses to the outbreak of COVID-19. We also found that the risk information proportion mediated the PTE effect, which provided evidence that the focusing illusion caused by agenda setting may be the psychological mechanism underlying the PTE effect, and it is likely a domain-general mechanism, which is worth conducting further investigations on in various major emergencies in the future. To mitigate the impact of PTE and avoid exaggerating estimates of the risks associated with the emergency, further study is needed to nudge the media to report objective, neutral, and comprehensive information about the risk source and ensure that the risk information received by audiences is proportional to the real-world risk.

## Figures and Tables

**Figure 1 ijerph-20-04350-f001:**
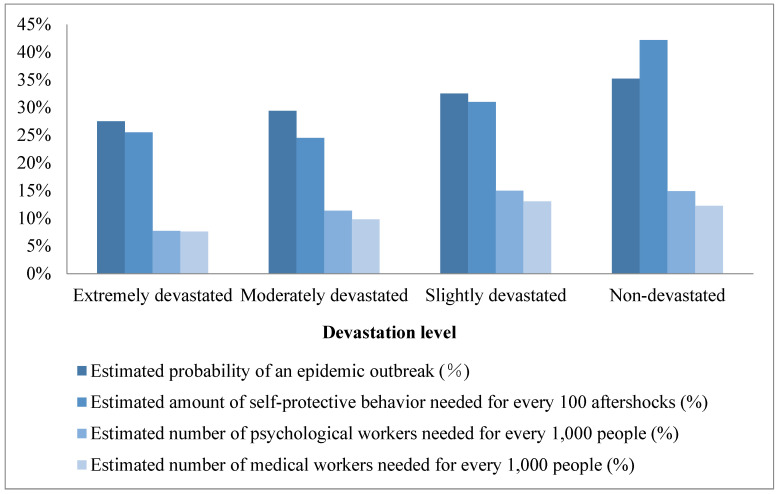
Residents’ post-earthquake concerns about safety and health issues in areas with varied devastation levels (replotted using the data of [4]).

**Figure 2 ijerph-20-04350-f002:**
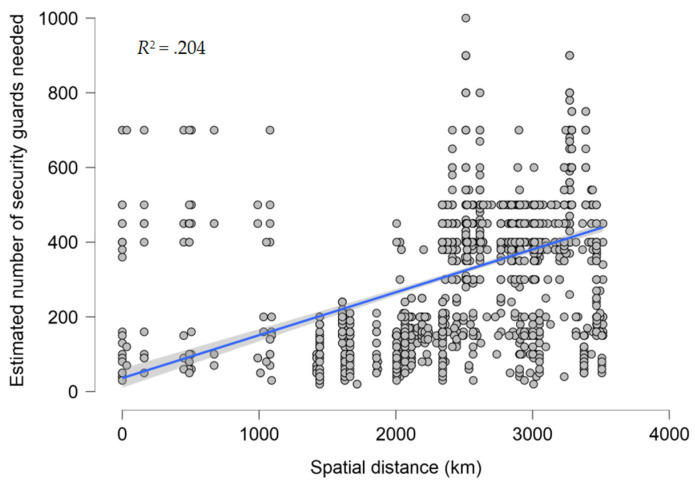
Scatterplot of spatial distance against the estimated number of security guards needed for every 10,000 participants in the China–Eurasia Expo, with the best-fitting regression line in the middle; *N* = 2034, *R*^2^ = .204 (replotted using the data of [13]).

**Figure 3 ijerph-20-04350-f003:**
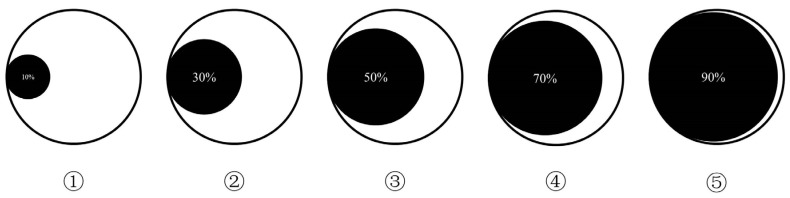
Diagram of risk information proportion. The outer circle represents the total information of the area where an emergency occurs. The inner black circle represents the risk-related (or hazard-related) information, and the blank part represents the information related to the area but nothing to do with the risk or hazard. ① indicates that the proportion of risk information is small, such as 10%, while ⑤ indicates that the proportion of risk information is large, such as 90%; ②, ③, and ④ represent different proportions of risk information between small and large, such as 30%, 50%, and 70%.

**Figure 4 ijerph-20-04350-f004:**
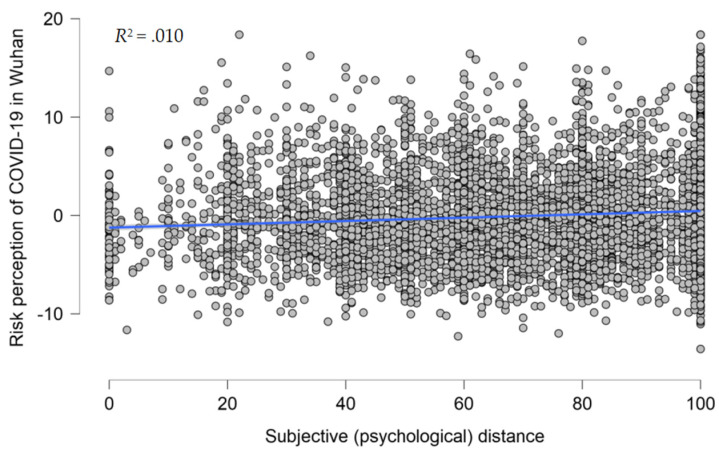
Scatterplot of subjective (psychological) distance against the risk perception of COVID-19 in Wuhan, with the best-fitting regression line in the middle; *N* = 7284, *r* = .098, *R*^2^ = .010.

**Figure 5 ijerph-20-04350-f005:**
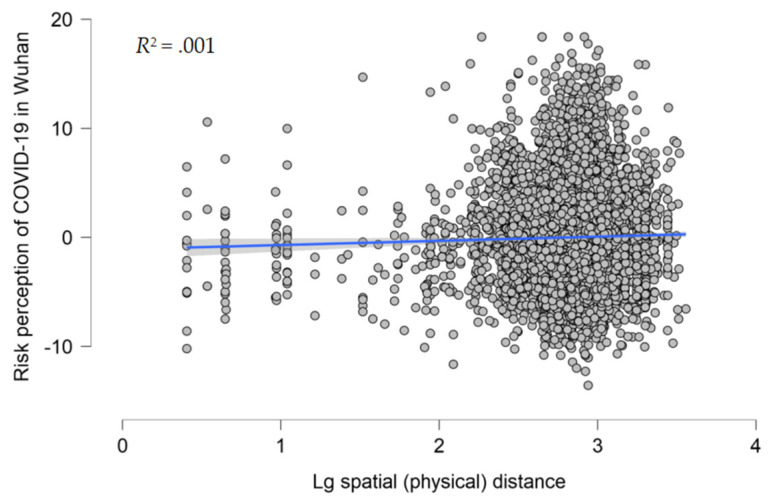
Scatterplot of the log transformation of spatial (physical) distance against the risk perception of COVID-19 in Wuhan, with the best-fitting regression line in the middle; *N* = 7283, *r* = .027, *R*^2^ = .001.

**Figure 6 ijerph-20-04350-f006:**
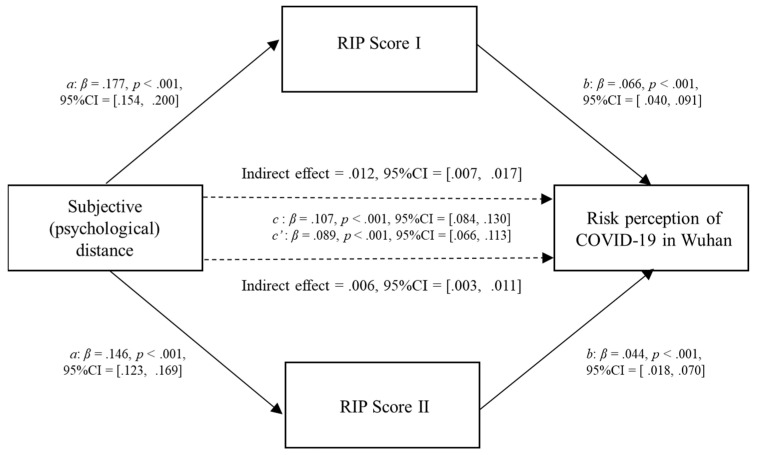
Parallel mediation model showing the effects of subjective (psychological) distance and risk information proportion on the risk perception of COVID-19 in Wuhan; *N* = 7284.

**Figure 7 ijerph-20-04350-f007:**
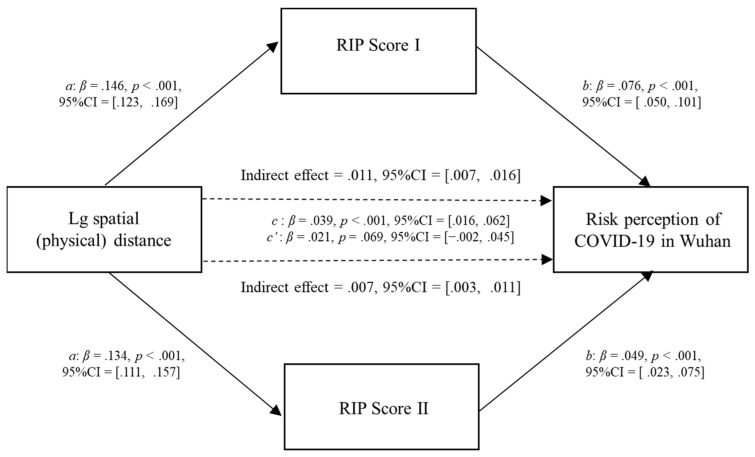
Parallel mediation model showing the effects of spatial (physical) distance and risk information proportion on the risk perception of COVID-19 in Wuhan; *N* = 7283.

**Figure 8 ijerph-20-04350-f008:**
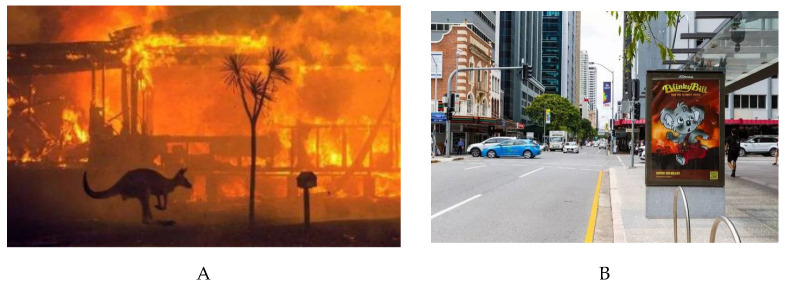
Screenshots of two ways of media reporting—taking the 2019–2020 fire season in eastern Australia as an example. The method shown in Screenshot (**A**) is a common way to report, in which the bushfire in Australia occupied the whole image area. However, Screenshot (**B**) utilizes a picture-in-picture technology that the proportion of the focus event in the whole picture has been adjusted; the warning sign about the Australian bushfire is displayed together with the surrounding peaceful streets.

**Table 1 ijerph-20-04350-t001:** Survey respondents’ demographic data (*N* = 8170).

.		Percentage			Percentage
**Gender**	Male	43.5	**Age**	Less than 17	2.7
	Female	56.5		18–20	13.9
**Identity**	1	68.3		21–30	45.6
	2	27.3		31–40	25.0
	3	2.3		41–50	9.2
	4	0.3		51–80	3.4
	5	0.2		81 and above	0.2
	6	1.7			

Note: Identity was coded as follows: 1 = people without contact with any COVID-19 cases, 2 = people who are aware of the confirmed cases in the neighborhood, 3 = medical staff and Centers for Disease Control (CDC) personnel involved in relevant cases, 4 = close contacts, 5 = confirmed cases or suspected cases, and 6 = others.

**Table 2 ijerph-20-04350-t002:** Items and factor loadings of the four-factor model.

Item	Description	Factor 1 (Personal Prevention)	Factor 2 (Healthcare)	Factor 3 (Environmental Isolation)	Factor 4 (Interpersonal Prevention)
13	Estimated number of cleaning frequently touched surfaces	.842			
12	Estimated body temperature measurement	.766			
14	Estimated number of clothing changes	.685			
1	Estimated number of medical doctors needed for every 1000 residents in Wuhan		.786		
2	Estimated number of psychological workers needed for every 1000 residents in Wuhan		.781		
10	Estimated days to abide by the quarantine rules			.587	
11	Estimated days of delaying school reopening			.585	
7	Estimated times of hand shaking				.572
5	Estimated number of social/family gatherings				.559
Proportion var. (%)		19.939	13.738	7.798	7.267
Cumulative (%) of variance explained		19.939	33.677	41.475	48.742

**Table 3 ijerph-20-04350-t003:** Means, SDs, and pairwise correlations of the measures (*N* = 7284, adult data with no missing values).

	Mean	SD	1	2	3	4	5	6	7	8	9	10	11
1 Gender	NA	NA	1										
2 Age	29.712	9.234	.156 ***	1									
3 Identity	NA	NA	−.004	−.026 *	1								
4 Importance	60.262	29.766	−.047 ***	.010	.053 ***	1							
5 Closeness	52.870	31.547	−.031 **	.055 ***	.112 ***	.609 ***	1						
6 Time exposed to news about COVID-19	4.362	1.317	−.001	.124 ***	.074 ***	.089 ***	.124 ***	1					
7 Subjective (psychological) distance	73.227	25.777	.061 ***	.146 ***	−.145 ***	−.001	−.074 ***	−.016	1				
8 Lg spatial (physical) distance	2.828	0.316	−.012	.060 ***	−.129 ***	−.093 ***	−.128 ***	−.026 *	.511 ***	1			
9 RIP Score I	0.804	0.125	−.031 **	.061 ***	−.129 ***	−.042 ***	−.071 ***	−.016	.197 ***	.169 ***	1		
10 RIP Score II	8.063	1.353	−.016	.094 ***	−.133 ***	−.061 ***	−.097 ***	−.001	.175 ***	.163 ***	.480 ***	1	
11 Risk perception of COVID-19 in Wuhan	0.000	4.473	−.100 ***	−.028 *	−.023 *	.156 ***	.086 ***	.099 ***	.098 ***	.027*	.098 ***	.080 ***	1

Note: M = mean; SD = standard deviation. Variables were coded as follows: gender: 0 = female, 1 = male; identity of participants: 0 = unrelated person, 1 = related person; time exposed to news about COVID-19: 1 = almost zero, 2 = less than 15 min, 3 = 15–30 min, 4 = 31–45 min, 5 = 46–60 min, 6 = more than one hour. * *p* < .05; ** *p* < .01; *** *p* < .001.

**Table 4 ijerph-20-04350-t004:** Hierarchical OLS regression of demographic variables and subjective (psychological) distance on the risk perception of COVID-19 in Wuhan (*N* = 7284, adult data with no missing values).

Variable	Model 1			Model 2		
	*B*	*SE*	β	*B*	*SE*	β
Step 1						
Gender	−0.815 ***	0.105	−.090	−0.852 ***	0.105	−.094
Age	−0.008	0.006	−.016	−0.015 **	0.006	−.031
Identity	−0.303 **	0.112	−.031	−0.167	0.112	−.017
Importance	0.024 ***	0.002	.159	0.023 ***	0.002	.151
Closeness	−0.001	0.002	−.009	0.000	0.002.	.002
Step 2						
Subjective (psychological) distance				0.018 ***	002	.106
*F* value		51.562 ***			56.914 ***	
*R* ^2^		.034			.045	
Adj. *R*^2^		.034			.044	
△ *R ^2^*					.011 ***	

* *p* < .05; ** *p* < .01; *** *p* < .001.

**Table 5 ijerph-20-04350-t005:** Hierarchical OLS regression of demographic variables and spatial (physical) distance on the risk perception of COVID-19 in Wuhan (*N* = 7283, adult data with no missing values).

Variable	Model 1			Model 2		
	*B*	*SE*	β	*B*	*SE*	β
Step 1						
Gender	−0.816 ***	0.105	−.090	−0.806 ***	0.105	−.089
Age	−0.008	0.006	−.016	−0.009	0.006	−.018
Identity	−0.303 **	0.112	−.031	−0.262 *	0.112	−.027
Importance	0.024 ***	0.002	.159	0.024 ***	0.002	.160
Closeness	−0.001	0.002	−.009	−0.001	0.002	−.005
Step 2						
Spatial (physical) distance				0.536 **	0.166	.038
*F* value		51.572 ***			44.781 ***	
*R* ^2^		.034			.036	
Adj. *R*^2^		.034			.035	
△ *R^2^*					.001 **	

* *p* < .05; ***p* < .01; ****p* < .001.

**Table 6 ijerph-20-04350-t006:** Parallel mediation model showing the effects of subjective (psychological) distance and risk information proportion on the risk perception of COVID-19 in Wuhan (*N* = 7284, adult data with no missing values).

	Mediator						Outcome Variable		
	RIP Score I			RIP Score II			Risk Perception of COVID-19 in Wuhan		
	*B*	*SE*	β	*B*	*SE*	β	*B*	*SE*	β
Subjective (psychological) distance	0.001 ***	0.000	.177	0.008 ***	0.001	.146	0.016 ***	0.002	.089
RIP Score I							2.352 ***	0.469	.066
RIP Score II							0.145 ***	0.043	.044
*F*		60.810 ***			60.418 ***			53.400 ***	
*R* ^2^		.055			.055			.062	

* *p* < .05; ** *p* < .01; *** *p* < .001.

**Table 7 ijerph-20-04350-t007:** Parallel mediation model showing the effects of spatial (physical) distance and risk information proportion on the risk perception of COVID-19 in Wuhan (*N* = 7283, adult data with no missing values).

	Mediator						Outcome Variable		
	RIP Score I			RIP Score II			Risk Perception of COVID-19 in Wuhan		
	*B*	*SE*	β	*B*	*SE*	β	*B*	*SE*	β
Spatial (physical) distance	0.058 ***	0.005	.146	0.573 ***	0.050	.134	0.302	0.166	.021
RIP Score I							2.706 ***	0.469	.076
RIP Score II							0.163 ***	0.044	.049
*F*		50.336 ***			57.182 ***			47.128 ***	
*R* ^2^		.046			.052			.055	

* *p* < .05; ** *p* < .01; *** *p* < .001.

## Data Availability

The datasets generated during and/or analyzed during the current study are available from the corresponding author on reasonable request.

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
