# Peer review of "An Agenda-Setting Account for Psychological Typhoon Eye Effect on Responses to the Outbreak of COVID-19 in Wuhan"

_ijerph, 2023, doi:10.3390/ijerph20054350_

Round 1

Reviewer 2 Report

According to the authors "... the practical implementation of a balanced agenda setting now appears to be constitutionally impossible," I would suggest therefore that the title of study emphasizes the discovery of the PTE effect. They further state that "the PTE effect detected so far is not a natural effect but man-made instead." They could then merely summarize or outline the conditions or factors and situations that make the PTE effect possible.     

Reviewer 3 Report

In the first paragraph, the PTE phenomena should be defined clearly, in what contexts has it been described, and how it is to be understood in the same context as presented in this study. If the context is not-matched, then it should be explained why this model is being used over others (i.e., concentric distance analyses).

The scatter plots through the manuscript should show the correlation coefficient and the effect size in the figure captions.

Data are otherwise sound.

However, the manuscript's references are thin and should be increased to a minimum of 70 as it does not show a large enough review of the literature. Additionally, more cross-cultural or other country comparisons could be made in the discussion and what findings can and cannot be generalized for external validity and why?

Reviewer 4 Report

Thank you for giving me the opportunity to review your manuscript and provide feedback on your work. The authors conducted a nationwide survey of 7,283 respondents from 31 provinces/municipalities in China to examine the relation between the distance to respondents city of residence from Wuhan and their safety concerns and risk perception of the epidemic that occurred in Wuhan City, which is a valuable work. The Authors have certainly spent some time reflecting upon the subject of this paper and has designed and implemented a promising project.

While the ambition of the paper is valuable, there are several issues that need to be seriously taken care of and that I would like to suggest to the authors for their future consideration.

The authors have mentioned several times that “We dubbed it the PTE effect in responses to the outbreak of COVID-19”. The authors have also made it clear that the PTE effect was first reported and named by Li et al. in their Wenchuan earthquake study. So Im wondering whether the authors are the first to use the the PTE effect to describe and explain the similar phenomena during the COVID-19 pandemic. Wang et al. (2020) have already investigated the Psychological Typhoon eye effect during the COVID-19 outbreak. Furthermore, Wen et al. (2020) have investigated the Psychological Typhoon Eye Effect and Ripple Effect in different COVID-19 severity regions.

The authors have missed out on one important element with regard to how it contributes to the existing literature. What exactly is the ultimate point of this work? What exactly is the targeted problem that this paper aims at proposing new ideas to deal with? How does such a work illustrate a problem that we have not realized or a new way of dealing with the Psychological Typhoon eye effect during the COVID-19 outbreak that previous scholarship could not accommodate?  

Most importantly, the authors seem to have already published several similar papers on this topic, e.g., Xu et al. 2020. So its important to declare the difference from the previous publication and whether they are using the same data.

Limitation or boundaries of the PTE effect and its comparison with other alternatives (e.g., the Ripple Effect) should be considered. For example, would the people living outside of China (e.g., the Americans) have the similar effect? Is this effect limited to a country? Are there similar studies in countries outside of China? Besides, is there any difference between people living in Hubei and other provinces? Will this effect still hold then?

“COVID-19 in Wuhan City” is not a suitable keyword.

Lines 29-30: The statement “The Coronavirus disease 2019 (COVID-19) epidemic first broke out in Wuhan in December 2019 and reached its peak in the city in February 2020” might be misleading about the origin of the Coronavirus.

Lines 39-40: “The PTE effect was first reported and named by Li et al. in their Wenchuan earthquake study.” Several citations should be added to refer to Li et al.

Line 131: change “working hypothesis” to “working hypotheses”

Do not include everything in the Introduction. Add a theoretical development section to specifically illustrate how these concepts have been connected with each other and how the hypotheses have been constructed and proposed. Move some paragraphs in the Introduction to this new section.

Line 142: “31provinces” “31 provinces”

For the survey respondents, is there a gap of the age group 81-99?

In this study, subjective (psychological) distance to Wuhan is measured by “how far away their residence is from Wuhan”. While the Spatial (physical) distance to Wuhan based on calculation of the authors might be more accurate, what is the essential difference between the psychological and physical distance? Some participants might just have a bad sense of spatial distance. What factors might influence peoples psychological distance to a place? How could these factor contribute to the current discussion or future research? It should be stated more clearly in the literature review or the Discussion section.

Sections 2.2.2. is too short and has the same title with section 2.2.3. .

Line 338: “Discussion and implication” “Discussion”

The copyrights issue of the two screenshots in Figure 8. Especially we can see that there is the WeChat official accounts name in the screenshot A.

Lines 404-411: The authors have mentioned correctly the debate on the of existence or naturalness of the PTE effect. But more discussion should be added and pointed out for future research.

Round 2

Reviewer 4 Report

The authors have done a good job in revising the manuscript.